# Human Umbilical Cord Mesenchymal Stem Cell-Derived Small Extracellular Vesicles Ameliorated Insulin Resistance in Type 2 Diabetes Mellitus Rats

**DOI:** 10.3390/pharmaceutics14030649

**Published:** 2022-03-16

**Authors:** Seng Kar Yap, Kian Leong Tan, Nor Yasmin Abd Rahaman, Nur Fazila Saulol Hamid, Der Jiun Ooi, Yin Sim Tor, Qi Hao Daniel Looi, Li Kar Stella Tan, Chee Wun How, Jhi Biau Foo

**Affiliations:** 1Ming Medical Sdn. Bhd., D3-3 (2nd Floor), Block D3, Dana 1 Commercial Centre, Jalan PJU 1A/46, Petaling Jaya 47301, Selangor, Malaysia; ysk_713@hotmail.com (S.K.Y.); looi_daniel@yahoo.com (Q.H.D.L.); 2School of Biosciences, Faculty of Health and Medical Sciences, Taylor’s University, Subang Jaya 47500, Selangor, Malaysia; kianleongtan96@gmail.com (K.L.T.); YinSim.Tor@taylors.edu.my (Y.S.T.); 3Department of Veterinary Pathology and Microbiology, Faculty of Veterinary Medicine, Universiti Putra Malaysia, Serdang 43400, Selangor, Malaysia; noryasmin@upm.edu.my (N.Y.A.R.); nurfazila@upm.edu.my (N.F.S.H.); 4Department of Oral Biology and Biomedical Sciences, Faculty of Dentistry, MAHSA University, Jenjarum 42610, Selangor, Malaysia; djooi@mahsa.edu.my; 5Centre for Drug Discovery and Molecular Pharmacology (CDDMP), Faculty of Health and Medical Sciences, Taylor’s University, Subang Jaya 47500, Selangor, Malaysia; 6My Cytohealth Sdn. Bhd., 18-2, Jalan Radin Bagus 1, Bandar Seri Petaling, Kuala Lumpur 57000, Malaysia; 7School of Pharmacy, Faculty of Health and Medical Sciences, Taylor’s University, Subang Jaya 47500, Selangor, Malaysia; stella11tan16@gmail.com; 8School of Pharmacy, Monash University Malaysia, Bandar Sunway 47500, Selangor, Malaysia; How.CheeWun@monash.edu

**Keywords:** small extracellular vesicle, exosomes, mesenchymal stem cell, type 2 diabetes mellitus, insulin resistance

## Abstract

Human umbilical cord mesenchymal stem cell-derived small extracellular vesicle (hUC-MSCs-sEVs) therapy has shown promising results to treat diabetes mellitus in preclinical studies. However, the dosage of MSCs-sEVs in animal studies, up to 10 mg/kg, was considered high and may be impractical for future clinical application. This study aims to investigate the efficacy of low-dose hUC-MSCs-sEVs treatment on human skeletal muscle cells (HSkMCs) and type 2 diabetes mellitus (T2DM) rats. Treatment with hUC-MSCs-sEVs up to 100 μg/mL for 48 h showed no significant cytotoxicity. Interestingly, 20 μg/mL of hUC-MSCs-sEVs-treated HSkMCs increased glucose uptake by 80–90% compared to untreated cells. The hUC-MSCs-sEVs treatment at 1 mg/kg improved glucose tolerance in T2DM rats and showed a protective effect on complete blood count. Moreover, an improvement in serum HbA1c was observed in diabetic rats treated with 0.5 and 1 mg/kg of hUC-MSCs-sEVs, and hUC-MSCs. The biochemical tests of hUC-MSCs-sEVs treatment groups showed no significant creatinine changes, elevated alanine aminotransferase (ALT) and alkaline phosphatase (ALP) levels compared to the normal group. Histological analysis revealed that hUC-MSCs-sEVs relieved the structural damage to the pancreas, kidney and liver. The findings suggest that hUC-MSCs-sEVs could ameliorate insulin resistance and exert protective effects on T2DM rats. Therefore, hUC-MSCs-sEVs could serve as a potential therapy for diabetes mellitus.

## 1. Introduction

Type 2 diabetes mellitus (T2DM) cases have been increasing over the years, and are now responsible for 90% of all diabetic patients [1]. In Malaysia, the overall prevalence of diabetes mellitus in adults increased in 2019. Ismail et al. [2] reported that around 3.9 million individuals were diagnosed with diabetes in 2019, 13.3% higher than the number of cases reported in 2015. T2DM is an abnormal glucose homeostasis condition with defective insulin secretion and action of insulin in the body. The persistent hyperglycemia can exhaust beta-cell function due to over-secretion of insulin, deteriorate insulin resistance, and eventually cause complications such as cardiovascular disease, dermopathy (skin disease), nephropathy (kidney damage), neuropathy (nerve damage) and retinopathy (vision loss).

Current anti-diabetic medications for T2DM treatment are accompanied by side effects and only temporarily control the blood glucose level to reduce the damage from complications [3,4,5]. Given these concerns, stem cell therapy has been suggested to repair the destroyed pancreatic β-cells and restore glucose homeostasis in diabetic patients, thereby curing diabetes [6]. Although mesenchymal stem cell (MSC)-based therapy has shown promising results in multiple studies, the potential tumorigenicity of MSCs remains a safety issue [7]. Furthermore, therapeutic applications of MSCs are limited by the efficacy of engraftment and the survival rate of transplanted MSCs. Within this frame, cell-free therapy has emerged as an alternative option that could resolve the safety concerns associated with the administration of living cells [8]. Paracrine factors secreted by MSCs have been reported as critical players in tissue repair and regeneration, and these signals are commonly transmitted through the extracellular vesicles (EVs) [9]. EVs are defined as nanoparticles naturally secreted by lipid bilayer cells that do not undergo replication [10]. EVs contain many vital mediators involved in either short or long-range intercellular communication. For instance, these mediators are proteins, lipid metabolites, DNA and RNA species [4,11,12]. Numerous reports revealed that the proteins and miRNAs carried by EVs are closely associated with both normal physiology and pathophysiology. Thus, the potential of EVs for various applications is now under the spotlight of the research community [4,12]. However, high dosage of MSCs-sEVs for clinical application is still a challenge due to the high cost and long harvesting time. The present study investigates the efficacy of low doses of human umbilical cord MSCs-sEVs treatment on improving insulin sensitivity and ameliorating insulin resistance of human skeletal muscle cells (HSkMCs) in vitro and in vivo in a T2DM rat model.

## 2. Materials and Methods

### 2.1. Materials and Reagents

Primary human skeletal muscle cells (HSkMCs) (ATCC PCS-950-01TM), Mesenchymal Stem Cell Basal Medium (PCS-500-030TM), Primary Skeletal Muscle Growth Kit (PCS-950-04TM) and Skeletal Muscle Differentiation Tool (PCS-950-050™) were purchased from the American Type and Culture Collection (ATCC), Manassas, Virginia, USA. Phosphate-buffered saline (PBS) was purchased from Gibco-Invitrogen Corporation, Carlsbad, CA, USA. Accutase, heparin solution and human platelet lysate (hPL) were purchased from STEMCELL^TM^ Technologies, Vancouver, Canada. Primary mouse antibodies anti-CD81 (B-11: sc-166029), anti-TSG101 (C-2: sc-7964), anti-GRP94 (H-10: sc-393402), streptozotocin (STZ) and insulin were purchased from Santa Cruz Biotechnology, Inc., Dallas, TX, USA. Ammonium molybdate and nicotinamide was purchased from Sigma-Aldrich, St. Louis, MO, USA. Tri-sodium citrate dihydrate, dimethyl sulfoxide (DMSO), paraformaldehyde and cytochalasin B (CB) were purchased from Nacalai Tesque, Inc., Kyoto, Japan. Glucose Uptake Assay Kit (ab136955) was purchased from Abcam, Cambridge, UK. Standard rat pellet feed was purchased from Gold Coin, Malaysia. Accu-Chek blood glucose test strips were purchased from Roche Diagnostics GmbH, Munich, Germany. Glibenclamide, citric acid, penicillin-streptomycin (Pen-Strep) and Dulbecco’s modified eagle medium (1.0g/L Glucose) with L- glutamine and sodium pyruvate (LG-DMEM) were purchased from Tokyo Chemical Industry Co, Tokyo, Japan. Rat INS (insulin) ELISA kit and rat HbA1c (glycosylated haemoglobin/haemoglobin A1c) ELISA kit were purchased from FineTest, Wuhan Fine Biotech Co., Wuhan, China. Ketamine and xylazine were purchased from Ilium Troy Laboratories, Australia.

### 2.2. Cell Culture of hUC-MSCs and Isolation of hUC-MSCs-sEVs

Human umbilical cord mesenchymal stem cells (hUC-MSCs) were purchased from Tissue Engineering Centre, UKM Medical Centre, Malaysia (Ethical approval number: UKM PPI/111/8/JEP-2019-608). hUC-MSC-derived small extracellular vesicles (hUC-MSCs-sEVs) were harvested from the cell culture medium as previously described by Tan et al. [13]. Briefly, when hUC-MSCs had reached 70–80% confluency, the medium was replaced with fresh LG-DMEM supplemented with 1% Pen-Strep (without hPL) as conditional medium (CM). After 48 h, the CM was collected and filtered through a 0.22 μm filter. The filtered CM was placed in a Millipore™ Centricon^®^ Plus-70 (100 kDa NMWL) and centrifuged at 3500× *g* for 15 min at 4 °C. The sample filter cup with a collection cup was then reverse centrifuged at 1000× *g* for 2 min at 4 °C to collect the hUC-MSCs-sEVs. To further purify the hUC-MSCs-sEVs, the collected solution was diluted with PBS (70× dilution) (Gibco, Grand Island, NY, USA) and centrifuged at 3500× *g* for 10 min at 4 °C. The sample filter cup with a collection cup was then reversed and centrifuged at 1000× *g* for 2 min at 4 °C to obtain the hUC-MSCs-sEVs. The concentration of the hUC-MSCs-sEVs sample was measured by using the bicinchoninic acid (BCA) assay and stored at −80 °C for further analysis.

### 2.3. Characterization of hUC-MSCs-sEVs

hUC-MSCs-sEVs were characterized as previously described [13]. Exosomal markers CD81 and TSG101, and the GRP94 negative control were determined using Western blot analysis. The morphology and particle size of hUC-MSCs-sEVs were identified using transmission electron microscopy (TEM) and nanoparticle tracking analysis (NTA).

For TEM, the diluted hUC-MSCs-sEVs sample was fixed with 0.1% paraformaldehyde (Nacalai Tesque, Kyoto, Japan) at room temperature for 30 min and then incubated on a carbon-coated copper grid. The grid was rinsed with ultrapure water and stained with 2% ammonium molybdate (Sigma-Aldrich, USA). After removing the excess solution, the grid was viewed using a Tecnai G2 F20 S-TWIN transmission electron microscope (FEI, USA). 

The NTA of diluted hUC-MSCs-sEVs was analysed with a Nanosight NS300 (Malvern Panalytical, UK) equipped with a CMOS camera, a 20 × objective lens, a blue laser module (488 nm) and NTA software v3.2. The samples of the diluted hUC-MSCs-sEVs were introduced into the NTA chamber using 1-mL disposable syringes. At a detection threshold of 5, the video for NTA measurements was recorded for 30 s at the equilibrated temperature of 25 °C.

### 2.4. Cell Culture of Human Skeletal Muscle Cells

Primary human skeletal muscle cells (HSkMCs) (ATCC PCS-950-01TM) were obtained from the American Type and Culture Collection (ATCC, USA). The cells were maintained in Mesenchymal Stem Cell Basal Medium (PCS-500-030TM, ATCC, USA) supplemented with Primary Skeletal Muscle Growth Kit (PCS-950-04TM, ATCC, USA). The cells were incubated in 37 °C of incubator supplemented with 5% CO_2_. The medium was refreshed every three days. To differentiate the cells into myotubes, the cells were cultured in Skeletal Muscle Differentiation Tool (PCS-950-050™, ATCC, Manassas, VA, USA) for 96 h. The differentiation medium was changed every 48 h. A cell passage number of <10 was adopted.

### 2.5. Cytotoxicity Test of hUC-MScs-sEVs on HSkMCs

MTT assay was performed to evaluate the safety profile of hUC-MSCs-sEVs against HSkMCs. Briefly, 5 × 10^3^ of HSkMCs were seeded in each well of 96-well plates and incubated overnight. The concentrations of hUC-MSCs-sEVs at 0, 6.25, 12.5, 25, 50 and 100 µg/mL were added into respective wells for 48 h incubation. At the end of the treatment period, each of the wells was incubated with 20 μL of 5 mg/mL MTT solution (Molekula, Darlington, UK) for 2 h. A 150 µL of DMSO (Nascalai Tesque, Kyoto, Japan) was added to each well to solubilise the purple formazan. The absorbance was measured at 570 nm excitation wavelength and 630 nm reference wavelength using the Biotek Epoch™ microplate reader (BioTek Instruments, Winooski, VT, USA). The cell viability was calculated for each well using the equation: Cell viability = (Average absorbance of treated cells/Average absorbance of control cells) × 100%. A graph of cell viability percentage versus treatments was plotted. Three independent assays with triplicate samples in each assay were conducted.

### 2.6. Glucose Uptake Assay

HSkMCs were seeded in a 96-well plate with a density of 5.0 × 10^4^ cells/cm^2^. After 24 h, the complete growth medium was replaced with a differentiation medium. The differentiation medium was replaced for every 48 h, and a minimum of 96 h was required for HSkMCs to differentiate into the fused multinucleated myotubes. The images were captured using a Nikon Eclipse Ti™ inverted light microscope (Nikon Instruments, NY, USA) to identify the development of multinucleated myotubes in HSkMCs.

Cytochalasin B (CB) is a cell-permeable mycotoxin that potently inhibits actin polymerisation, yet CB is often used for glucose transporters (GLUT) 1-4 inhibition assay [14,15,16]. In this study, CB (NacalaiTesque, Kyoto, Japan) was used as a glucose transporter inhibitor to inhibit 2-DG uptake in the HSkMCs. The reduction of GLUT receptors for 2-DG binding could mimic an insulin resistance of the T2DM in vitro model. The glucose uptake assay was performed following the manufacturer’s instructions using a Glucose Uptake Assay Kit (ab136955, Abcam, Cambridge, UK). For the non-CB-treated cells experiment, HSkMCs were seeded onto the 96-well plate and divided into 4 groups, as shown in Table 1.

All groups were starved in serum-free LG-DMEM (Tokyo Chemical Industry Co., Tokyo, Japan) overnight (<16 h). Simultaneously, the hUC-MSCs-sEVs treatment group was added with 20 μg/mL of hUC-MSCs-sEVs during the overnight serum starvation. After serum starvation, all groups were washed thrice with PBS and incubated with 2% BSA (Nacalai Tesque, Kyoto, Japan) in Kreb’s buffer for 40 min. The EVs-stimulated HSkMCs group was treated with an additional 20 μg/mL hUC-MSC-sEVs during the incubation. After incubation, 1 μM of insulin (Santa Cruz Biotechnology, Dallas, TX, USA) was added to the insulin-stimulated HSkMCs group and incubated for 30 min. All groups were then added with 10 µL of 10 mM 2-DG, except for the HSkMCs-only group. All groups were incubated for 30 s. The solutions from all groups were aspired and discarded. All groups were washed thrice with PBS. The cells were lysed using the extraction buffer and then neutralised with neutralising buffer. The samples were centrifuged at 200 × *g* for 5 min to collect the supernatants. The supernatants and standards were added to a new 96-well plate. Each well was added with 10 μL of reaction mix A (a mixture of assay buffer and enzyme mix in 4:1 ratio) and incubated for 1 h at 37 °C, followed by incubation with extraction buffer at 90 °C for 40 min. Next, the wells were cooled down on ice for 5 min. The samples were neutralised with neutralising buffer and treated with 38 μL of reaction mix B (a mixture of glutathione reductase, substrate and recycling mix in a 10:8:1 ratio). The absorbance was obtained every 2–3 min on the Biotek Epoch™ microplate reader (BioTek Instruments, VT, USA) with a wavelength of 412 nm in kinetic mode at 37 °C. Three independent assays with triplicate samples in each assay were conducted.

HSkMCs were seeded onto a 96-well plate and divided into 5 groups for the CB-treated cells experiment, as shown in Table 2. The procedure was conducted as described in the previous process of glucose uptake assay for non-CB-treated cells. The 30 min incubation with insulin was changed to 30 min incubation with both 1 μM of insulin and 0.5 µM of CB.

### 2.7. Animal Study

The use of animals in the present study was approved by the Institutional Animal Care and Use Committee, Universiti Putra Malaysia (project approval number UPM/IACUC/AUP-R047/2020). Experimental procedures and handling of the animals were performed as stipulated by the Universiti Putra Malaysia Code of Practice for the Care and Use of Animals for Scientific Purposes. Male Sprague Dawley rats (*n* = 28, 8-week-old, 200 ± 20 g) were used in the present study. The rats were acclimatised for one week at a temperature of 25 ± 2 °C and a 12/12 h light/dark cycle. Animals were housed under standard conditions with food and water given ad libitum.

### 2.8. Treatment of Experimentally STZ Induced Diabetic Rats

Following acclimatisation, the normal control rats were fed a standard pellet diet, while the other rats were fed a high-fat diet (HFD) for seven weeks. The high-fat diet cake was made weekly with the composition of 50% ground standard pellet, 20% refined cooking oil, 20% full cream milk powder, 5% refined sugar, 2.5% starch and 2.5% cholesterol. The body weight of all rats was measured every week. The non-fasting HFD-fed rats were administered nicotinamide (120 mg/kg body weight) (Sigma-Aldrich, MO, USA) and STZ (60 mg/kg body weight) (Santa Cruz Biotechnology, TX, USA) via intraperitoneal (IP) injection at a 15 min interval. The STZ solution was prepared freshly and administered within 5 min upon dissolution. The standard pellet diet-fed rats were employed as a control reference, and received 200 µL of PBS and 0.05 M sodium citrate buffer (pH 4.5), replacing nicotinamide and STZ, respectively. After 7 days, the rats were fasted for 6 h (8:00 a.m. to 2:00 p.m.). The fasting blood glucose levels were measured using OneTouch UltraEasy^®^ Glucometer (Roche Diagnostics GmbH, Munich, Germany) to confirm the development of diabetes mellitus. The remaining HFD-fed rats with unsuccessful diabetes development were further overnight fasted and injected with a second dose of STZ (40 mg/kg) via IP injection. After 3 days, these rats were fasted for 6 h (8:00 a.m. to 2:00 p.m.). The fasting blood glucose levels were measured using OneTouch UltraEasy^®^ Glucometer. The HFD-fed rats with high fasting blood glucose levels (>15 mmol/L) were used as the diabetic rats in the present study.

T2DM rats were randomly divided into the following groups (*n* = 4/group), and the treatment intervention, as stated in Table 3, was performed for 15 days. The rats, except the normal control group, were remained on a high-fat diet during the treatment period. Upon the successful induction of the T2DM model, the diabetic rats were treated with hUC-MSCs-sEVs (every 3 days, total 5 doses) and hUC-MSCs (every 2 weeks, total 2 doses) via tail vein intravenous injection, adopted from Sun et al. [17] and Si et al. [18] respectively.

### 2.9. Oral Glucose Tolerance Test

An oral glucose tolerance test (OGTT) was performed on the next day of the last treatment intervention. After overnight fasting, each rat received one dose of dextrose (2 g/kg body weight) via oral gavage. Blood glucose levels were determined at the set intervals of 0, 30, 60 and 120 min using the One Touch UltraEasy^®^ Glucometer (Roche Diagnostics GmbH, Munich, Germany). A graph of the blood glucose concentrations following oral glucose loads against time was plotted. The areas under the curves (AUCs) of the chart for 0–30 min and 0–120 min were computed geometrically.

### 2.10. Histological Analysis

On the following day of OGTT, the overnight fasted rats were euthanised using a combination of 100 mg/kg ketamine (Ilium Troy Laboratories, Australia) and 10 mg/kg xylazine (Ilium Troy Laboratories, Australia). Whole blood samples were collected via cardiac puncture. The pancreas, liver and kidney were harvested and stored in 4% paraformaldehyde (Nacalai Tesque, Kyoto, Japan). The organ samples were sent to the Veterinary Laboratory Services Unit, Faculty of Veterinary Medicine, Universiti Putra Malaysia, for histological analysis.

### 2.11. Analysis of Blood and Serum Parameters

Whole blood samples were sent to the Veterinary Laboratory Services Unit, Faculty of Veterinary Medicine, Universiti Putra Malaysia, for the complete blood count and biochemistry tests. Fasting serum insulin and fasting serum HbA1c of the rats were quantitated using the Rat INS (Insulin) and rat HbA1c (glycosylated haemoglobin/haemoglobin A1c) enzyme-linked immunosorbent assay (ELISA) kits (FineTest, Wuhan, China) respectively, following the manufacturer’s instructions. Quantitative insulin sensitivity check index (QUICKI), homeostatic model assessment of insulin resistance (HOMA-IR) and β-cell function (HOMA-β) were computed according to the formula below:
QUICKI = 1/(log fasting blood glucose (mmol/L) + log fasting serum insulin (µU/mL))HOMA-IR = (fasting blood glucose (mmol/L) × fasting serum insulin (µU/mL))/22.5HOMA-β = (20 × fasting serum insulin (µU/mL))/(fasting blood glucose (mmol/L) − 3.5)

### 2.12. Statistical Analysis

All results were expressed as mean ± standard deviation (SD). The statistical significance of differences was determined by using SPSS version 28.0 statistic software (SPSS Inc., Chicago, IL, USA). In vitro study results were analysed by one-way ANOVA with Tukey’s post hoc test. In vivo study results were analysed by one-way ANOVA with the least significant difference (LSD) post hoc test. A *p*-value less than 0.05 (*p* < 0.05) was considered statistically significant.

## 3. Results

### 3.1. Characterization of hUC-MScs-sEVs

The micrographs of the hUC-MSCs-sEVs from transmission electron microscopy (TEM) displayed a bilayer membrane of cup-like morphology with a diameter around 100 nm (Figure 1A,B). The Western blot analysis of hUC-MSCs-sEVs showed positive bands for exosomal markers CD81, TSG101 and negative for GRP94 (Figure 1C). GRP94 is an endoplasmic reticulum marker that is absent in sEVs. It should be noted that the GRP94 expression was detected in cell lysate but not in sEVs. NTA showed that hUC-MSCs-sEVs have a mode of particle size at 97 nm, while other peaks of particle size were 74, 149 and 209 nm. Overall, the particle size of sEVs was less than 220 nm (Figure 1D). The protein concentration of our hUC-MSCs-sEVs was approximately 5 mg/mL.

### 3.2. CytotoxicityTest of hUC-MScs-sEVs on HSkMCs

MTT assay was used to investigate the cytotoxic effect of hUC-MSCs-sEVs on HSkMCs. In Figure 2, all concentrations of hUC-MSCs-sEVs treatment groups showed a significantly higher percentage of viability (*p* < 0.05) than untreated-HSkMCs for 48 h. The viability of hUC-MSCs-sEVs-treated HSkMCs ranged from 120.6 ± 7.5% to 148.0 ± 17.8%. The highest percentage of cell viability was observed at 100 µg/mL of the hUC-MSCs-sEVs treatment group.

### 3.3. Glucose Uptake Assay

HSkMCs were differentiated according to ATCC guidelines in 96-well plates before conducting the glucose uptake assay. The images of undifferentiated (Figure 3A) and differentiated HSkMCs (Figure 3B,C) were observed under inverted light microscopy. The differentiation of HSkMCs into myotube cells was observed. The myotubes cells showed multiple nuclei in the cytoplasm.

The glucose uptake in the insulin-stimulated group was significantly increased, by 1.5 times compared to the non-insulin-stimulated group, while the hUC-MSCs-sEVs treated group significantly improved 2-DG uptake (*p* < 0.01) compared with the non-insulin stimulated group by 1.8 times (Figure 4A). Interestingly, the hUC-MSCs-sEVs treatment group showed a significantly higher 2-DG uptake activity (*p* < 0.05) than the insulin-stimulated group. CB treatment showed significant inhibition of 2-DG uptake activity on insulin-stimulated HSkMCs by 4 times (*p* < 0.001) compared to non-CB-treated insulin-stimulated HSkMCs (Figure 4B). The addition of 20 µg/mL hUC-MSCs-sEVs into the CB-treated insulin-stimulated cells did not significantly improve (*p* > 0.05) the 2-DG uptake activity.

### 3.4. STZ Induced T2DM in High Fat Diet Rat Model

The blood glucose level of rats was monitored weekly. STZ is transported into pancreatic beta cells through GLUT 2 and causes beta cell damage, whereas nicotinamide is administered to partially protect the rat’s beta cells against STZ [19]. A week after the first dose injection of STZ and nicotinamide, one rat was confirmed with hyperglycaemia (fasting blood glucose > 15 mmol/L). After 3 days of the second injection of low dose STZ (40 mg/kg), the bedding in the cages of the STZ-induced rats was abnormally wet, causing us to suspect the development of diabetes in the rats. After blood glucose level measurement, the successful diabetes induction rate was 94%, with fasting blood glucose > 15 mmol/L. The diabetic rats showed symptoms of diabetes such as thirst (higher water consumption than usual), hyper-urination (wet bedding) and weight loss (Figure 5).

In addition, the body weight of rats was measured weekly over the study (Figure 5). The body weight of all groups of rats increased steadily over the time before STZ injection, where no significant difference (*p* > 0.05) in body weight was observed before the 8^th^ week. Once the diabetic model was confirmed during the 8th week, all diabetic rats showed significant body weight loss compared to normal rats, reducing by approximately 15% during the 9th and 10th weeks. The body weights of the treatment groups were decreasing with a similar trend to the untreated diabetes group, suggesting that the weight loss was due to diabetes mellitus.

### 3.5. Oral Glucose Tolerance Test

OGTT was carried out at the end of treatment to evaluate the effects of hUC-MSCs-sEVs and hUC-MSCs treatments on the ability of diabetic rats to respond to a glucose disposal challenge. All groups showed a sharp increase at the peak of 30 min after the oral glucose challenge, except that the normal group showed stable blood glucose levels at below 7.0 within 0 to 120 min (Figure 6A). The first 30 min of OGTT results were plotted to investigate the glucose tolerance during the first 30 min interval (Figure 6B). The gradient of the normal group was 0.05 ± 0.03. However, the untreated diabetes group showed a large spike in blood glucose level with a gradient of 0.42 ± 0.08 compared to the normal group, indicating a poor glucose tolerance. The glibenclamide-treated group significantly suppressed the gradient by twofold, with a value of 0.21 ± 0.07 compared to the untreated diabetes group. hUC-MSCs-sEVs and hUC-MSCs groups showed an improvement in glycaemic response during the 30-min. hUC-MSCs-sEVs at 0.25, 0.5 and 1 mg/kg significantly reduced the gradient (*p* < 0.05) to 0.25 ± 0.08, 0.25 ± 0.05 and 0.3 ± 0.01 respectively, compared to the untreated diabetes group. The hUC-MSCs-treated group also significantly reduced the gradient (*p* < 0.05) to 0.3 ± 0.06 compared to the untreated diabetes group. The total area under the curve (AUC) of the glycaemic response in OGTT was calculated for each treatment group (Figure 6C). The AUC of the normal group was 12.33 ± 1.85. However, the untreated diabetes group showed a larger AUC with a value of 53.44 ± 3.18 compared to the normal group, indicating a poor glycaemic control. The glibenclamide-treated and hUC-MSCs-treated groups showed significantly smaller AUCs (*p* < 0.05) with values of 45.9 ± 7.25 and 47.95 ± 4.08, respectively, compared to the untreated diabetes group. The hUC-MSCs-sEVs treatment groups at 0.25, 0.5 and 1 mg/kg also showed significantly smaller AUCs (*p* < 0.05) with 45.72 ± 1.65, 47.74 ± 3.81 and 44.34 ± 2.96, respectively, compared to the untreated diabetes group.

### 3.6. Analysis of Blood and serum Parameters

The quantitative insulin sensitivity check index (QUICKI) value of the normal group was significantly higher (0.51 ± 0.01) compared to all diabetic groups indicating an excellent insulin sensitivity in the rats (Figure 7A). The induction of diabetes significantly reduced the QUICKI value to 0.39 ± 0.02. Meanwhile, glibenclamide treatment significantly increased the QUICKI value to 0.42 ± 0.01 compared to the untreated diabetes group. Among the hUC-MSCs-sEVs-treated groups, the 1 mg/kg dose had a significantly higher QUICKI value (0.42 ± 0.02) than the untreated diabetes group. The remaining concentration of the hUC-MSCs-sEVs and the hUC-MSC-treated group did not significantly differ from the untreated diabetes group.

Homeostatic model assessment of insulin resistance (HOMA-IR) value was calculated to interpret the effect of treatments on insulin resistance of the diabetic animal groups (Figure 7B). The normal group possessed the lowest HOMA-IR value (4.19 ± 0.45), showing low insulin resistance. In the untreated diabetic rats, the HOMA-IR value was significantly increased to 17.99 ± 5.28, indicating a high insulin resistance state. Compared to the untreated diabetes group, glibenclamide and 1 mg/kg of hUC-MSCs-sEVs significantly reduced the HOMA-IR value by 39% and 35% to 10.97 ± 2.05 and 11.69 ± 2.56, respectively. However, no significant difference was found in 0.25 and 0.5 mg/kg of hUC-MSCs-sEVs treatment groups and hUC-MSCs treatment group compared to the untreated diabetes group.

The normal group possessed the highest HOMA-β value (166.3 ± 52.6), suggesting normal and functional pancreatic β-cells in the rats (Figure 7C). In the untreated diabetes rats, the HOMA-β value was reduced to 31.1 ± 2.8, indicating damaged pancreatic function due to STZ. No significant difference in HOMA-β value was observed in 0.25 and 1 mg/kg body weight of the hUC-MSCs-sEVs-treatment group and hUC-MSCs-treated group compared to the untreated diabetic group. However, glibenclamide and 0.5 mg/kg of hUC-MSCs-sEVs showed slightly lower HOMA-β values of 22.1 ± 5.7 and 21.5 ± 4.9, respectively, compared to the untreated diabetes group.

Fasting serum glycated haemoglobin (HbA1c) was examined at the end of treatment (Figure 7D). The untreated diabetes group showed significantly higher fasting serum HbA1c at 76.2 ± 11.0 ng/mL (*p* < 0.05) compared to hUC-MSCs-sEVs treatment at 0.5 and 1 mg/kg, hUC-MSCs and normal groups, while it showed no significant difference compared to the glibenclamide and 0.25 mg/kg of hUC-MSCs-sEVs treatment groups.

The complete blood count of untreated diabetic rats showed no significant difference in the mean corpuscular volume (MCV), mean corpuscular haemoglobin concentration (MCHC), band neutrophils (Band N), neutrophils and basophils when compared to non-diabetic rats (Table 4). Red blood cells (RBC), haemoglobin (Hb), packed cell volume (PCV), white blood cells (WBC), lymphocytes, monocytes, eosinophils and platelets (PLT) in the untreated diabetes group showed significantly lower amounts (*p* < 0.05) compared to the non-diabetes group. Interestingly for RBC, Hb and MCV, both glibenclamide and 1 mg/kg of hUC-MSCs-sEVs treatment groups showed no significant differences compared to the normal group. The treatments of both hUC-MSCs-sEVs at 1 mg/kg and glibenclamide groups showed no significant difference in WBC, neutrophils, lymphocytes, monocytes, eosinophils and basophils compared to the normal group. In contrast, the untreated diabetic and hUC-MSCs-treated groups showed significantly lower values of WBC, lymphocytes, monocytes, eosinophils, and PLT than the non-diabetic animal group.

### 3.7. Kidney and Liver Functions

The kidney function of untreated diabetic rats showed no significant difference in the creatinine (Creat) compared to normal group. Sodium (Na), potassium (K), chloride (Cl) and urea in the untreated diabetes group were significantly higher (*p* < 0.05) compared to the normal group (Table 5). Interestingly, hUC-MSCs-sEVs and hUC-MSCs treatment groups showed no significant differences in the Na, K, Cl and Creat compared to normal rats, except the 0.5 mg/kg hUC-MSCs-sEVs group only showed a significantly lower value (*p* < 0.05) in Cl compared to the normal group. Only the hUC-MSCs-sEVs and hUC-MSCs treatment groups showed significantly higher urea values (*p* < 0.05) than the normal group.

For liver function, the untreated diabetic rats showed no significant difference in the total protein (TP), albumin (ALB), globulin (Glo), albumin and globulin (A:G) ratio, aspartate aminotransferase (AST), direct bilirubin (DBil) and total bilirubin (TBil) when compared to normal group. All diabetic rats among treatments and untreated groups showed significantly higher alkaline phosphatase (ALP) and alanine aminotransferase (ALT) than the non-diabetic group.

### 3.8. Histological Analysis

Pancreas sections of non-diabetic rats showed the normal architecture of the pancreatic islet (Figure 8A). However, the untreated diabetic rats showed pathological changes of Langerhans islets which were deranged and shrunken. Additionally, Inflammation was found in the untreated diabetic rats with evidence of leukocytic cells infiltration (yellow arrow as shown in Figure 8B). The glibenclamide treatment group showed hydropic degeneration of endocrine cells, which regenerated toward their normal configuration. The treatment group of hUC-MSCs, and 0.25 and 1 mg/kg hUC-MSCs-sEVs restored pancreatic islet architecture. However, the 0.5 mg/kg hUC-MSCs-sEVs treatment group showed degranulation of the cytoplasm with hydropic degeneration of necrotic hepatocytes.

The hepatocytes from the rat liver sections of non-diabetic rats appeared normal and showed no significant change in hepatic architecture (Figure 9A). Untreated diabetic rats showed obvious macrovesicular steatosis with multiple round vacuoles indicating numerous small to medium-sized fat droplets of fatty cells within the liver parenchymal. The treatment groups of glibenclamide and hUC-MSCs-sEVs at 0.25 and 1 mg/kg showed the restoration of normal liver architecture with no specific pathology or inflammation compared to untreated diabetic rats. The hUC-MSCs treatment group showed minimal scattered individual fat cells. However, 0.5 mg/kg of the hUC-MSCs-sEVs treatment group showed a foamy appearance of hepatocytes and was filled with numerous small lipid vacuoles.

The non-diabetic animal group showed normal histological structure of the glomerulus within kidney tissue (Figure 10A). All hUC-MSCs-sEVs treatments and glibenclamide-treated animal groups showed restoration of kidney architecture and appeared normal basement membrane without necrotic cells. In contrast, the untreated diabetes group showed glomerulosclerosis and inflammation of the glomerulus by infiltrating with the inflammatory cells and thickening of the basement membrane of the Bowman’s capsule (red arrow in Figure 10B) and widening of the Bowman’s space (* in Figure 10B). Morevoer, the epithelial lining of proximal convoluted tubules exhibited cytoplasmic vacuolation (blue arrow in Figure 10B), and the kidney section showed extravasation of blood with oedema (black arrow in Figure 10B) in the untreated diabetic rats. Some tubules show rupture of their lining epithelium and desquamation of the cells into the lumen (green arrow in Figure 10B). Similarly, the kidney section of the hUC-MSCs treatment group showed vacuolation of proximal convoluted tubules, desquamation of the cells into the lumen and inflammation cells infiltration.

## 4. Discussion

In the present study, the characterisation results of hUC-MSCs-sEVs indicated the successful isolation before the experimental study (Figure 1). The treatment with hUC-MSCs-sEVs promoted the growth of HSkMCs (Figure 2), implying that hUC-MSCs-sEVs are safe to be employed as therapeutic agents up to 100 μg/mL. Interestingly, the hUC-MSCs-sEVs were also found to promote glucose uptake in HSkMCs (Figure 4A). After that, cytochalasin B (CB) was employed to inhibit GLUT4 to mimic T2DM [15,16]. However, glucose uptake was not promoted by hUC-MSCs-sEVs in the CB-treated cells (Figure 4B). This could be due to the CB being a potent glucose transporter inhibitor. The concentration of hUC-MSCs-sEVs (20 µg/mL) used for the CB-treated cells may not be sufficient to overcome the action of CB. Despite the negative results from the hUC-MSCs-sEVs treatment on CB-treated cells, our findings suggested that hUC-MSCs-sEVs may promote glucose uptake via glucose transporters 1 to 4 (GLUT1-4). We acquired the T2DM rat model for further investigation to determine the efficacy of hUC-MSCs-sEVs on improving insulin resistance and insulin sensitivity in diabetic rats.

Based on our critical review of the study by Sun et al. [17], a treatment dose of 10 mg/kg was utilised in the animal study. Our earlier benchmarking for the other MSCs-sEVs regenerative studies showed that most of the treatment dosage ranges were less than 1 mg/kg to promote therapeutic effect. This suggests that the dosage of 10 mg/kg could be too high and may affect the cost and sEV harvesting time during a clinical application. This suggestion was further supported by the findings in the present study, which showed hUC-MSCs-sEVs at 1 mg/kg or lower could effectively improve glucose tolerance in the T2DM animal model. This information is crucial to reduce the treatment cost and the harvesting time for future clinical application.

At the end of treatment, an oral glucose tolerance test (OGTT) for 2 h was conducted on all the overnight-fasting rats (Figure 6). At the first 30 min of OGTT (Figure 6B), all groups showed a spike in blood glucose level compared to the non-diabetic animal group. In contrast, the negative group showed the steepest gradient of increase among the other groups. All the treatments of hUC-MSCs-sEVs-treated and hUC-MSCs-treated groups showed significant differences (*p* < 0.05) compared to the untreated diabetes group. The finding reveals that the sEVs and hUC-MSCs treatment groups had better a glucose tolerance function that could suppress the blood glucose level after glucose intake for 30 min. This further confirms the potential of hUC-MSCs-sEVs as an adjunctive treatment for T2MD. As expected, the hUC-MSCs-treated group showed a similar effect to the hUC-MSCs-sEVs-treated group. This may indicate that the isolated hUC-MSCs-sEVs carry the components of mediators from hUC-MSCs that improve glucose tolerance in the T2DM animal model. Upon calculation of AUC of OGTT at 2 h (Figure 6C), we found a significantly higher AUC (*p* < 0.05) for the untreated diabetis group compared to the glibenclamide, hUC-MSCs-sEVs and hUC-MSCs treatment groups. Correspondingly, this shows that the hUC-MSCs-sEVs treatment could improve the glucose tolerance function of T2DM rats, similar to hUC-MSCs and anti-diabetic drug glibenclamide treatments.

The findings from the blood serum analysis of HbA1c (Figure 7D) reveal that the treatment groups improve the hyperglycemia of diabetic rats by lowering the glycation of haemoglobin, except glibenclamide and 0.25 mg/kg of hUC-MSCs-sEVs treatment groups. This result goes against glibenclamide’s therapeutic efficacy, which should diminish the percentage of glycation of haemoglobin. We speculate that the severity of diabetes for the glibenclamide treatment group was exceptionally high in fasting blood glucose (>20 mmol/L). The condition of severe damage to beta cells would compromise the drug’s efficacy. A study by Sokolovska et al. [20] found that glibenclamide treatment given at 2 mg/kg for 6 weeks on STZ-induced diabetic rats with blood glucose (>13.89 mmol/L) showed no effect on the change of glycated haemoglobin level, suggested to be due to the loss of a large number of beta cells after STZ administration. Another study by Ooi et al. [21] showed that glibenclamide was given at 10 mg/kg for 32 days on STZ-induced diabetic rats with blood glucose (9 ± 2 mmol/L) which displayed significantly lower glycated haemoglobin level. We suggest that glibenclamide treatment would give a positive response on the glycated haemoglobin, in which mildly diabetic rats with low fasting blood glucose (<11 mmol/L). The finding reveals that 0.5 and 1 mg/kg of hUC-MSCs-sEVs and hUC-MSCs treatment would improve diabetes mellitus by lowering the glycated haemoglobin level in T2DM rats.

Further analysing the insulin resistance and insulin sensitivity in the T2DM rat model, we obtained the fasting serum insulin and fasting blood glucose to compute the HOMA-IR, HOMA-β and QUICKI indexes (Figure 7). In the present study, we found that solely analysing fasting serum insulin could not reflect the actual state of insulin secretion. Some may argue that the fasting serum insulin level may indicate the function of the pancreas response to blood glucose homeostasis. However, this absolute value of fasting serum insulin may not reflect the actual state of insulin secretion during food intake. Therefore, it is recommended to measure the blood serum insulin levels during OGTT, which could reflect the actual state of insulin secretion during food intake. However, collecting blood from rats during OGTT may pose a challenge since a substantial amount of blood is needed for the analysis. Both glibenclamide-treated and the highest concentration hUC-MSCs-sEVs-treated groups (1 mg/kg) improved QUICKI values indicating both treatments enhanced the insulin sensitivity in diabetic rats (Figure 7A). Interestingly, the HOMA-IR values in both glibenclamide-treated and hUC-MSCs-sEVs-treated group (1 mg/kg) showed lower values (*p* < 0.05) than the untreated diabetes group (Figure 7B). However, HOMA-β values were not improved among all the treatment groups (Figure 7C). This finding suggests that treatment with these concentrations of hUC-MSCs-sEVs and hUC-MSCs may not be able to regenerate and repair the damaged pancreatic islets in diabetic rats. However, it is still too early to conclude that a low dose of hUC-MSCs-sEVs could not restore the damaged pancreas. According to Sun et al. [17], the concentration of hUC-MSCs-sEVs at 10 mg/kg could relieve the destroyed islets in T2DM rats by inhibiting STZ-induced apoptosis. Therefore, we conducted a histological analysis to examine the pancreatic islets further. As such, histological examination of pancreatic islets is crucial to confirm the potential regeneration of damaged pancreas by hUC-MSCs-sEVs.

A histological examination of pancreas sections of the untreated diabetic rats showed pathological changes of Langerhans islets which were deranged and shrunken (Figure 8). The glibenclamide and 0.25 and 1 mg/kg of hUC-MSCs-sEVs treatment groups showed protection or restoration of pancreatic islet architecture compared to the untreated diabetes group. The result is in agreement with a study by Sun et al. [17] who indicated that hUC-MSC derived exosomes could relieve pancreatic destruction and promote islet regeneration in STZ-induced diabetic rats. These findings suggest hUC-MSCs-sEVs treatment could exert a protective or regenerative effect on the diabetic rat’s pancreas against the damage from diabetes mellitus.

Other than the pancreas, we also examined the kidney and liver because they are the target organs of diabetic complications. At the same time, we investigated the regenerative or cytotoxic effect of hUC-MSCs-sEVs treatment groups on the kidneys and liver of diabetic rats. For the kidney function test (Table 5), the blood urea and creatinine (Creat) levels are the markers of nephrotoxicity implicated in the diagnosis of kidney damage [22,23,24]. Our findings are consistent with Balamash et al. [22], who found no significant increase in serum creatinine in the diabetic control group compared to the normal control group. In contrast, the level of blood urea nitrogen (BUN) in the diabetic animal group was significantly higher when compared to the normal group [22]. However, another study by Sun et al. [17] showed no significant differences in BUN and Creat between hUC-MSC-derived exosomes-treated diabetic groups and a normal group, indicating the exosomes were unlikely to have damaged kidney tissue. We speculate that the unchanged Creat and elevated blood urea levels of diabetic rats in the present study could be a result of the high-fat diet.

A further histological analysis was conducted to determine the severity of kidney damage. The haematoxylin and eosin (H-E)-stained tissue section of the untreated diabetes group presented a rupture of the lining of proximal convoluted tubules and cytoplasmic vacuolation in the kidney tissue (Figure 10). These histological changes observed in the untreated diabetic group agree with those reported previously by Balamash et al. [22]. In the present study, all the hUC-MSCs-sEVs treatment groups and the glibenclamide-treated animal group appeared to have restored kidney architecture and normal basement membrane with an absence of necrotic cells. The result suggests that the protective effect of hUC-MSCs-sEVs treatment may be attributble to the restoration of the normal architecture of the kidney in STZ-induced diabetic rats, thereby exhibiting a protective role against renal damage.

For the liver function test, the untreated and untreated diabetic groups showed no significant differences in TP, ALB, Glo, A:G ratio, AST, DBil and TBil compared to the non-diabetic group (Table 5). All diabetic rats among all the treatment groups and the untreated group showed significantly higher activities of both ALP and ALT compared to the non-diabetes group. Our result is supported by a previous study that reported increased ALP and ALT activities in experimentally STZ-induced diabetic rats but no changes in TP, ALB and bilirubin activities [22]. ALP is a potent anti-inflammatory mediator that protects tissues from damage. The increased ALP activity in blood serum could result from diabetes-induced damage to liver tissue. As biomarkers of hepatocellular injury, AST and ALT are less specific to the liver due to their presence in skeletal muscle, and the enzyme activity could be elevated from skeletal muscle injury [25]. The findings suggest all treatment groups showed no improvement in the elevated ALP and ALT activities of liver function in diabetic rats.

To further assess the severity of liver damage, the histological architecture of hepatocytes from liver sections of each group was observed (Figure 9). The treatment groups of 0.25 and 1 mg/kg hUC-MSCs-sEVs and glibenclamide had restored normal liver architecture with no specific pathology or inflammation compared to untreated diabetic rats. This finding agrees with a previous study by Al-ani et al. [26], who found glibenclamide-treated STZ-induced diabetic rats showed fewer pathological changes to liver architecture, indicating a protective effect of the treatment against hepatic changes associated with diabetes. In addition, the finding is supported by Diaz-Juarez et al. [27], who found that elevated hepatic enzymes did not correlate with cell necrosis. Surprisingly, the study showed that a partial hepatectomy in rat liver induced higher ALT and AST serum levels that did not affect the liver regenerative capacity. The present study’s finding suggests that hUC-MSCs-sEVs treatments at 0.25 and 1 mg/kg could exert a protective or regenerative effect on the livers of diabetic rats without causing further cytotoxicity in the liver tissue.

Complete blood counts of the untreated diabetes group and hUC-MSCs treatment group showed lower RBC, Hb, and PCV levels than the non-diabetic animal group. This finding is in line with a study by Erukainure et al. [28], who indicated that RBC, Hb and PCV were found to have lower values in diabetic rats and suggested the occurrence of anaemia (Table 4). In the present study, the 1 mg/kg of hUC-MSCs-sEVs and glibenclamide treatment groups showed no alterations of RBC and Hb compared to the non-diabetic group. Thus, the result reveals that the treatment of hUC-MSCs-sEVs at 1 mg/kg showed a protective or stimulation effect on RBC and Hb levels, improving the anaemia-like condition in diabetic rats. Moreover, reductions in the concentrations of WBC, lymphocytes, monocytes and PLT in untreated diabetic rats indicate suppression of the immune system. These blood cells are responsible for invading pathogens by releasing complements or phagocytosis. The reductions in these blood cells agree with a study by Erukainure et al. [28], who found the levels of WBC, PLT and lymphocytes were reduced in diabetic rats. This condition could lead to various complications associated with diabetes mellitus. In this study, both glibenclamide and 1 mg/kg of hUC-MSCs-sEVs treatment of diabetic rats showed no significant difference in WBC, neutrophils, lymphocytes, monocyte, eosinophils and basophils compared to the non-diabetic animal group. This finding reveals that the treatment of hUC-MSCs-sEVs at 1 mg/kg could enhance the immune system of diabetic rats, which is crucial in reducing infection and increasing survival rate.

## 5. Conclusions

The present study represents a novel report on low dose hUC-MSCs-sEVs to promote glucose uptake in HSkMCs and improve insulin tolerance in diabetic rats by lowering HOMA-IR and suppressing blood glucose spikes. Histological analysis demonstrated the protective or regenerative effects of 1 mg/kg of hUC-MSCs-sEVs on the architecture of the pancreatic islets, hepatocytes and renal glomeruli. The kidney- and liver function tests for the hUC-MSCs-sEVs treated diabetic rats revealed that the sEVs are unlikely to damage kidney and liver tissues. Moreover, blood profile analysis showed that hUC-MSCs-sEVs treatment could be beneficial for diabetic rats to recover from the anaemia-like symptoms and increase immunity by improving the erythrocytes and haemoglobin levels as well as maintaining the number of white blood cells. Overall, the present study’s findings could provide valuable information to develop hUC-MSCs-sEVs as a novel therapy to alleviate diabetes mellitus by restoring glucose homeostasis in diabetic patients. Significantly, the low dose hUC-MSCs-sEVs treatment could reduce the harvesting time and the cost of production in the future clinical applications without compromising the efficacy of the treatment.

## Figures and Tables

**Figure 1 pharmaceutics-14-00649-f001:**
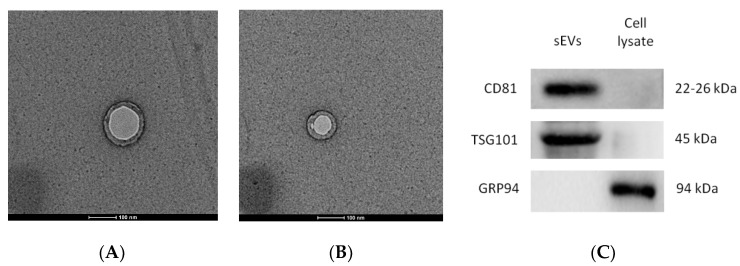
Validation of hUC-MSCs-sEVs (**A**) and TEM micrographs of hUC-MSCs-sEVs (**B**). Exosomal markers were detected by Western blotting (**C**). Nanoparticle size analysis (**D**). The scale bars for (**A**,**B**) represent 100 nm.

**Figure 2 pharmaceutics-14-00649-f002:**
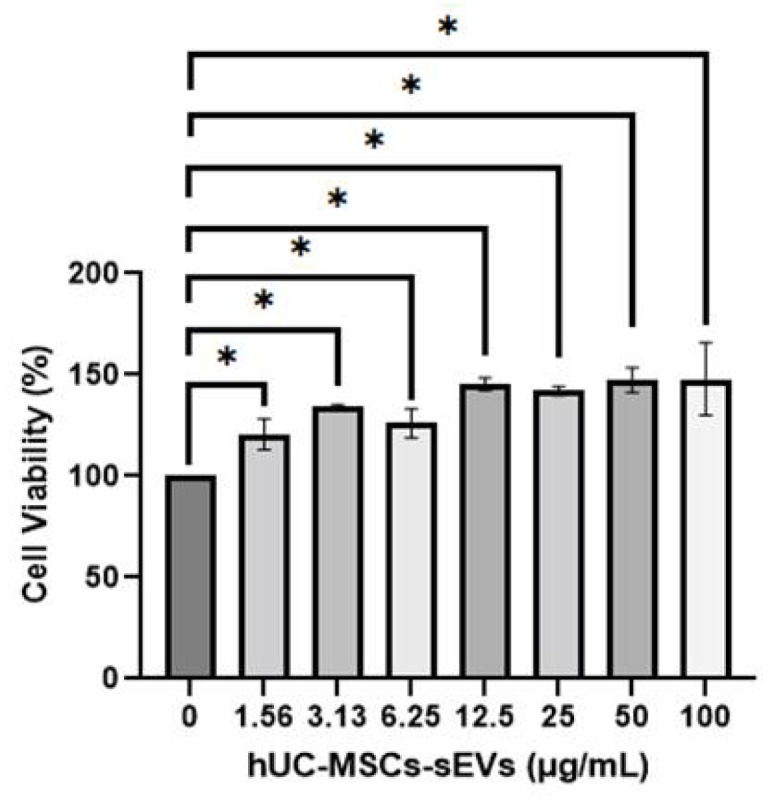
Cell viability of HSkMCs after being treated with different concentrations of hUC-MSCs-sEVs for 48 h. hUC-MSCs-sEVs showed no toxicity effect towards HSkMCs. Data are expressed as the mean ± SD of at least three independent assays with triplicate samples in each assay (* *p* < 0.05, one-way ANOVA with Tukey post hoc tests).

**Figure 3 pharmaceutics-14-00649-f003:**
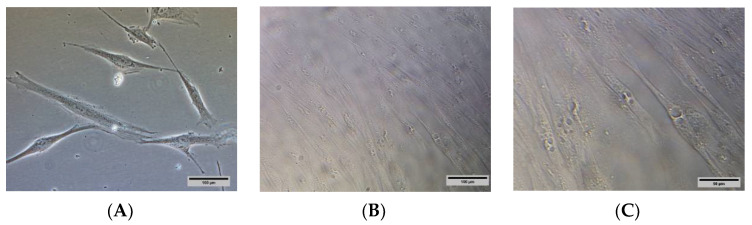
(**A**) Undifferentiated HSkMCs and (**B**) differentiated HSkMCs into fused multinucleated myotubes (yellow arrows). (**C**) Enlarged image of (**B**), scale bar represents 50 µm. Images were captured under Nikon Eclipse Ti™ inverted light microscopy. (**A**,**B**): 200× magnification, scale bar represents 100 µm.

**Figure 4 pharmaceutics-14-00649-f004:**
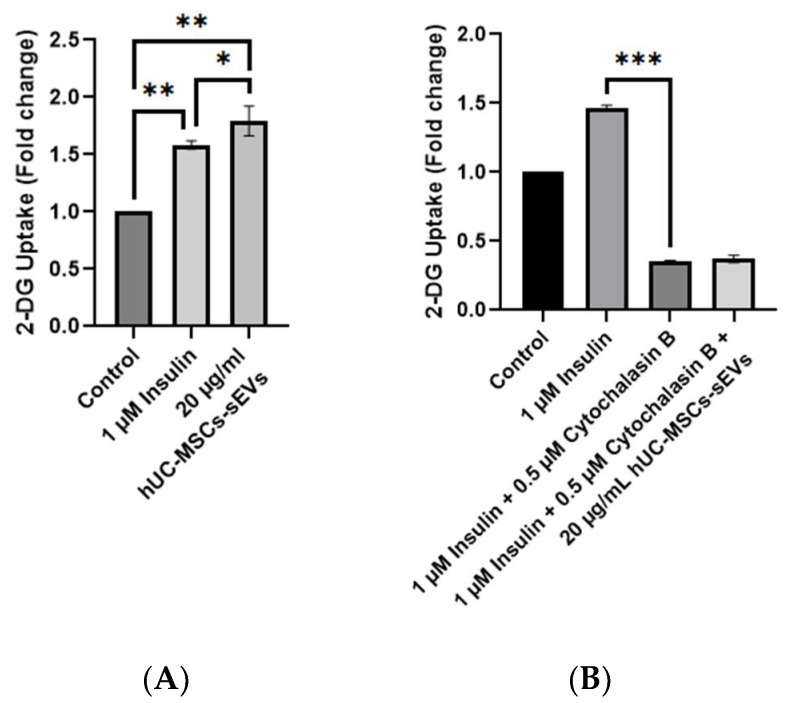
2-DG uptake by HSkMCs with and without CB treatment. (**A**) Comparison of insulin and hUC-MSCs-sEVs on stimulation of 2-DG uptake by HSkMCs. Treatment of hUC-MSCs-sEVs at 20 µg/mL showed a significant increase in 2-DG uptake by HSkMCs. (**B**) Effect of hUC-MSCs-sEVs on 2-DG uptake by cytochalasin B-treated HSkMCs. Cytochalasin B treatment at 0.5 µM significantly reduced the 2-DG uptake in HSkMCs. However, hUC-MSCs-sEVs seems unable to reverse the action of cytochalasin B. Data are expressed as the mean ± SD of at least three independent assays with triplicate samples in each assay (* *p* < 0.05, ** *p* < 0.01, *** *p* < 0.001, one-way ANOVA with Tukey post hoc tests).

**Figure 5 pharmaceutics-14-00649-f005:**
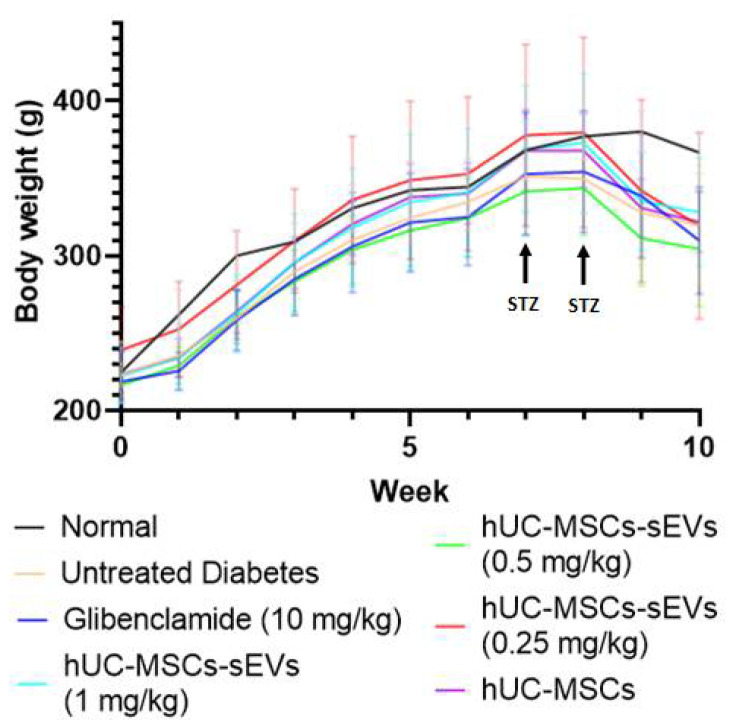
Weekly body weight changes of all groups of Sprague Dawley rats. Data are expressed as the mean ± SD (*n* = 4, *p* < 0.05, one-way ANOVA with LSD post hoc tests).

**Figure 6 pharmaceutics-14-00649-f006:**
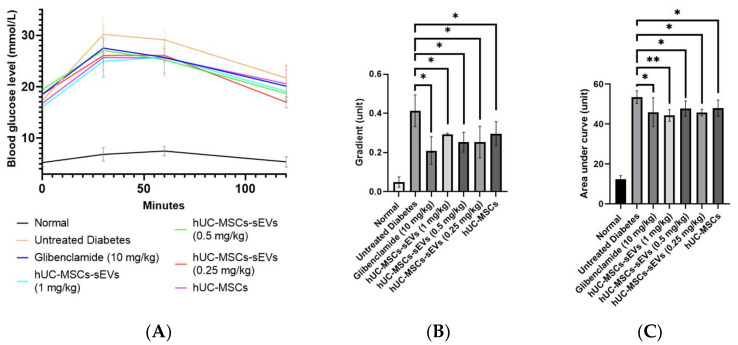
Effect of treatments on diabetic rats for oral glucose tolerance test between 0–120 min. (**A**) Blood glucose levels of rats for each interval of OGTT at 0, 30, 60 and 120 min. (**B**) Gradients at first 30 min of OGTT for different treatment groups. All treatment groups showed significantly lower gradient units compared to untreated diabetic rats. (**C**) The area under the curve of OGTT (0–120 min) for different treatment animal groups. All treatment groups showed a significantly lower area under the curve (AUG) than untreated diabetic rats. Data are expressed as the mean ± SD (*n* = 4, * *p* < 0.05, ** *p* < 0.01, one-way ANOVA with LSD post hoc tests).

**Figure 7 pharmaceutics-14-00649-f007:**
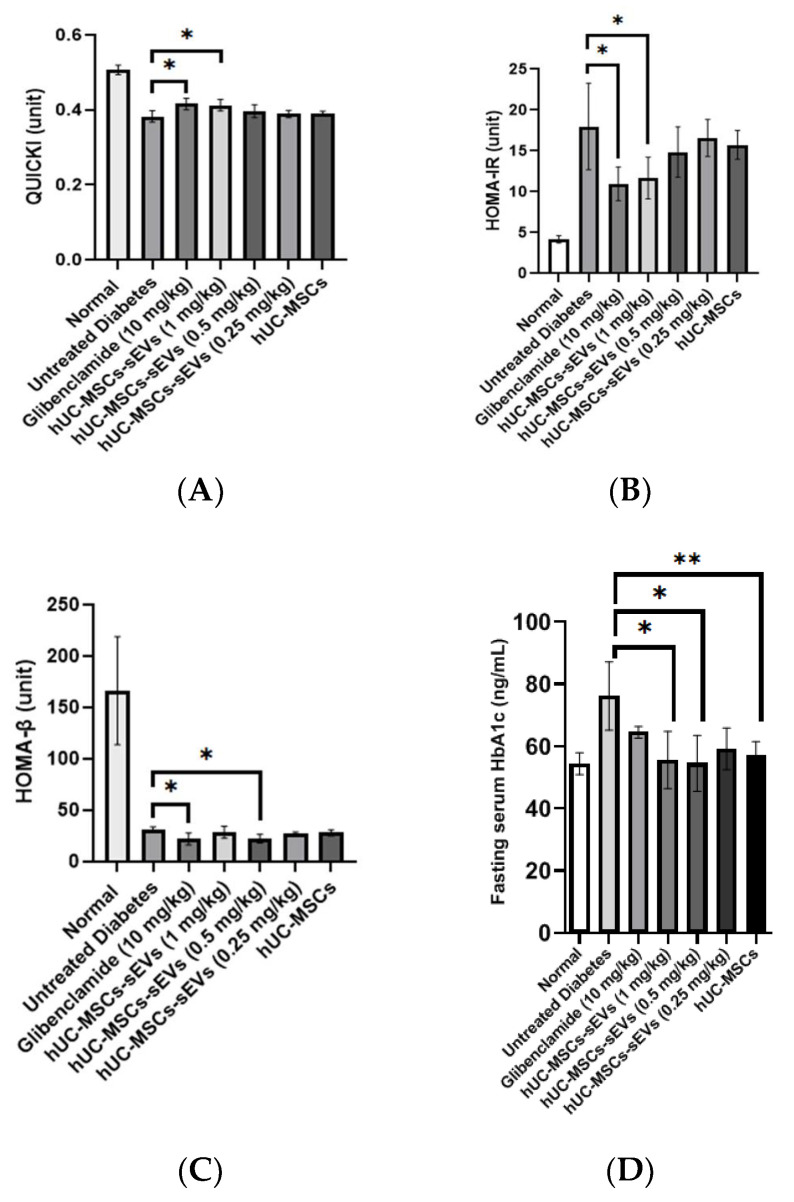
(**A**) QUICKI values of different treatment groups. The glibenclamide-treated and 1 mg/kg hUC-MSCs-sEVs-treated diabetic rats showed significantly higher QUICKI values than untreated diabetic rats. (**B**) HOMA-IR values of different treatment animal groups. The glibenclamide-treated and 1 mg/kg hUC-MSCs-sEVs-treated diabetic rats showed significantly lower HOMA-IR values than untreated diabetic rats. (**C**) HOMA-β values of different treatment groups. The glibenclamide-treated and 0.5 mg/kg hUC-MSCs-sEVs-treated diabetic rats showed significantly lower HOMA-β values than untreated diabetic rats. (**D**) Effect of different treatments on fasting serum HbA1b level of T2DM rats. Treatment of 0.5 and 1 mg/kg of hUC-MSCs-sEVs and hUC-MSCs showed significantly lower serum HbA1c than the untreated diabetes group. Data are expressed as the mean ± SD (*n* = 4, * *p* < 0.05, ** *p* < 0.01, one-way ANOVA with LSD post hoc tests).

**Figure 8 pharmaceutics-14-00649-f008:**
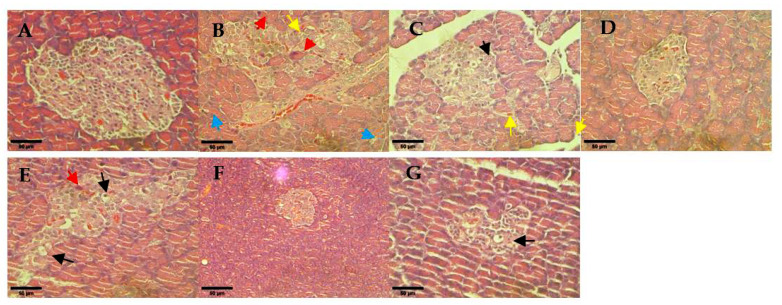
Histological observation of rat pancreas for different treatment diabetes groups: (**A**) non-diabetes; (**B**) untreated diabetes; (**C**) glibenclamide; (**D**) 0.25 mg/kg hUC-MSCs-sEVs; (**E**) 0.5 mg/kg hUC-MSCs-sEVs; (**F**) 1 mg/kg hUC-MSCs-sEVs; (**G**) hUC-MSCs. The red arrows show cell necrosis. The yellow arrows show leukocytic cell infiltration. The blue arrows show cytoplasmic vacuolation. The black arrows show hydropic degeneration. All sections were stained with haematoxylin and eosin and viewed under an inverted light microscope. 400× magnification; Scale bar represents 50 µm.

**Figure 9 pharmaceutics-14-00649-f009:**
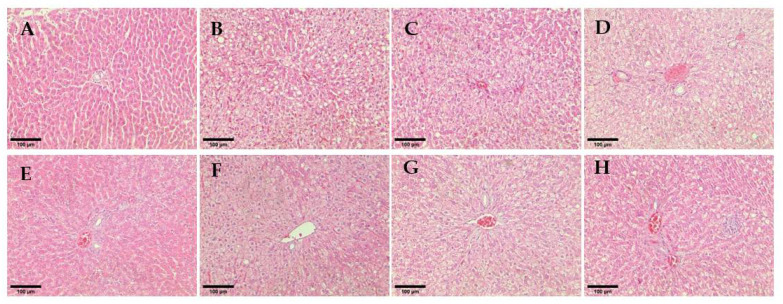
Histological observation of rat hepatocytes for different treatment groups. (**A**) Non-diabetes; (**B**) Untreated diabetes; (**C**) Glibenclamide; (**D**,**E**) 0.25 mg/kg hUC-MSCs-sEVs; (**F**) 0.5 mg/kg hUC-MSCs-sEVs; (**G**) 1 mg/kg hUC-MSCs-sEVs; (**H**) hUC-MSCs. All sections were stained with haematoxylin and eosin and viewed under an inverted light microscope. 200× magnification; Scale bar represents 100 µm.

**Figure 10 pharmaceutics-14-00649-f010:**
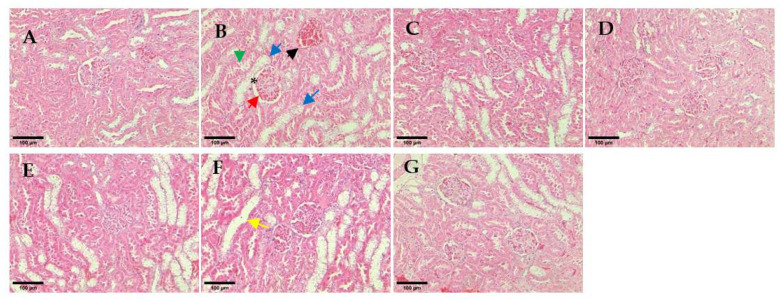
Histological observation of renal glomeruli. (**A**) Non-diabetes; (**B**) untreated diabetes; (**C**) glibenclamide; (**D**) 0.25 mg/kg hUC-MSCs-sEVs; (**E**) 0.5 mg/kg hUC-MSCs-sEVs; (**F**) 1 mg/kg hUC-MSCs-sEVs; (**G**) hUC-MSCs. The yellow arrow shows a dilated tubule due to loss of tubular lining. The red arrow shows the thickening of the basement membrane of the Bowman’s capsule. The green arrow shows a rupture of the lining epithelium and desquamation of the cells into the lumen. The blue arrows show cytoplasmic vacuolations and oedematous changes. The black arrow shows extravasation of blood with oedema. * shows thickening of the basement membrane of the Bowman’s capsule. All sections were stained with haematoxylin and eosin and viewed under an inverted light microscope. 200× magnification; Scale bar represents 100 µm.

**Table 1 pharmaceutics-14-00649-t001:** Treatment of hUC-MSCs-sEVs on HSkMCs.

Group	Description	2-DG (10 mM)
1	HSkMCs only (Blank)	No
2	HSkMCs only (Normal control)	10 μL
3	HSkMCs + 1 μM Insulin (Positive control)	10 μL
4	HSkMCs + 20 μg/mL hUC-MSC-sEVs	10 μL

**Table 2 pharmaceutics-14-00649-t002:** Treatment of hUC-MSCs-sEVs on CB-treated HSkMCs.

Group	Description	2-DG (10 mM)
1	HSkMCs only (Blank)	No
2	HSkMCs only (Normal control)	10 μL
3	HSkMCs + 1 μM Insulin (Positive control)	10 μL
4	HSkMCs + 0.5 μM CB + 1 μM Insulin	10 μL
5	HSkMCs + 0.5 μM CB + 1 μM Insulin + 20 μg/mL hUC-MSC-sEVs	10 μL

**Table 3 pharmaceutics-14-00649-t003:** Animal treatment plan.

Animal Group	Description	Treatment	Route	Frequency
NC	Normal control -normal pellet diet-fed -non-diabetic rats	PBS	IV injection via tail vein	Every 3 days, total 5 times
DM	Diabetic control without medication -HFD-fed -STZ-induced diabetes	PBS	IV injection via tail vein	Every 3 days, total 5 times
DM + GC	Diabetic control with glibenclamide drug -HFD-fed -STZ-induced diabetes	Glibenclamide (10 mg/kg body weight)	Oral gavage	Every day
DM + 1MG EV	Diabetic test rats -HFD-fed -STZ-induced diabetes	hUC-MSC-sEVs (1 mg/kg body weight)	IV injection via tail vein	Every 3 days, total 5 times
DM + 0.5MG EV	Diabetic test rats -HFD-fed -STZ-induced diabetes	hUC-MSC-sEVs (0.5 mg/kg body weight)	IV injection via tail vein	Every 3 days, total 5 times
DM + 0.25MG EV	Diabetic test rats -HFD-fed -STZ-induced diabetes	hUC-MSC-sEVs (0.25 mg/kg body weight)	IV injection via tail vein	Every 3 days, total 5 times
DM + MSC	Diabetic test rats -HFD-fed -STZ-induced diabetes	hUC-MSCs (2 million cells)	IV injection via tail vein	Every two weeks, total 2 times

**Table 4 pharmaceutics-14-00649-t004:** Complete blood counts of different groups at the end of treatment.

	Normal Control	Untreated Diabetes	Glibenclamide (10 mg/kg)	hUC-MSCs-sEVs (1 mg/kg)	hUC-MSCs-sEVs (0.5 mg/kg)	hUC-MSCs-sEVs (0.25 mg/kg)	hUC-MSCs (2 Million Cells)
RBC (×10^12^/L)	8.97 ± 0.71 ^a^	7.40 ± 0.64 ^b,c^	8.65 ± 1.02 ^a,c^	8.04 ± 0.80 ^a,b^	7.03 ± 1.90 ^b,c^	7.91 ± 1.00 ^a,b^	6.32 ± 0.72 ^b^
Hb (g/L)	157.80 ± 14.34 ^a^	121.20 ± 13.41 ^b^	144.67 ± 15.04 ^a,b^	142.40 ± 16.94 ^a,b^	129.50 ± 13.44 ^a,b^	120.75 ± 25.9 ^b^	114.50 ± 6.36 ^b^
PCV (L/L)	0.45 ± 0.04 ^a^	0.38 ± 0.02 ^b^	0.44 ± 0.05 ^a^	0.39 ± 0.02 ^b^	0.39 ± 0.03 ^a,b^	0.36 ± 0.04 ^b,c^	0.30 ± 0.03 ^c^
MCV (fL)	50.14 ± 1.30 ^a^	48.59 ± 3.11 ^a^	50.92 ± 2.5 ^a^	46.28 ± 4.45 ^a^	47.77 ± 2.86 ^a^	45.9 ± 0.73 ^a^	47.52 ± 0.95 ^a^
MCHC (g/L)	350.86 ± 3.79 ^a,b^	336.13 ± 15.49 ^a,b^	329.23 ± 7.25 ^b^	375.58 ± 21.44 ^c^	332.71 ± 7.57 ^a,b^	362.98 ± 8.88 ^a,c^	382.37 ± 14.84 ^c^
WBC (×10^9^/L)	14.13 ± 4.21 ^a^	7.60 ± 2.79 ^b^	13.87 ± 1.56 ^a^	12.88 ± 4.00 ^a^	9.45 ± 1.77 ^a,b^	6.13 ± 1.18 ^b^	5.15 ± 1.63 ^b^
Band N (×10^9^/L)	0.14 ± 0.04 ^a^	0.08 ± 0.03 ^a^	0.14 ± 0.02 ^a^	0.13 ± 0.04 ^a^	0.09 ± 0.02 ^a^	0.06 ± 0.01 ^a^	0.05 ± 0.02 ^a^
%	1.00 ± 0.00 ^a^	0.80 ± 0.45 ^a^	1.00 ± 0.00 ^a^	1.00 ± 0.00 ^a^	1.00 ± 0.00 ^a^	0.75 ± 0.50 ^a^	1.00 ± 0.00 ^a^
Neutro (×10^9^/L)	2.54 ± 0.54 ^a^	1.40 ± 0.32 ^a^	2.23 ± 0.43 ^a^	2.70 ± 1.05 ^a^	2.05 ± 1.04 ^a^	1.66 ± 0.54 ^a^	2.07 ± 1.15 ^a^
%	18.50 ± 3.00 ^a^	16.00 ± 1.63 ^a^	16.00 ± 2.00 ^a^	18.00 ± 5.29 ^a^	21.00 ± 7.07 ^a^	26.67 ± 4.16 ^a^	46.00 ± 36.77 ^a^
Lymp (×10^9^/L)	10.27 ± 3.46 ^a^	5.80 ± 2.02 ^b^	10.14 ± 0.71 ^a^	8.65 ± 3.13 ^a,b^	5.07 ± 2.39 ^b^	4.06 ± 0.55 ^b^	2.75 ± 2.53 ^b^
%	72.00 ± 3.65 ^a^	76.75 ± 1.71 ^a^	73.33 ± 3.06 ^a^	76.00 ± 3.74 ^a^	71.33 ± 6.11 ^a^	66.67 ± 5.03 ^a^	48.00 ± 33.94 ^b^
Mono (×10^9^/L)	0.90 ± 0.36 ^a^	0.42 ± 0.36 ^b^	1.22 ± 0.56 ^a^	0.68 ± 0.36 ^a,b^	0.55 ± 0.43 ^a,b^	0.23 ± 0.17 ^b^	0.25 ± 0.18 ^b^
%	7.40 ± 0.89 ^a^	5.40 ± 2.97 ^a^	8.67 ± 3.06 ^a^	5.80 ± 2.28 ^a^	7.00 ± 4.36 ^a^	4.00 ± 1.63 ^a^	4.50 ± 2.12 ^a^
Eosin (×10^9^/L)	0.12 ± 0.05 ^a^	0.00 ± 0.00 ^b^	0.14 ± 0.02 ^a^	0.1 ± 0.07 ^a^	0.11 ± 0.09 ^a^	0.05 ± 0.03 ^b^	0.03 ± 0.04 ^b^
%	1.00 ± 0.00 ^a^	0.00 ± 0.00 ^b^	1.00 ± 0.00 ^a^	0.80 ± 0.45 ^a^	1.33 ± 0.58 ^a^	0.75 ± 0.50 ^a^	0.50 ± 0.71 ^a,b^
Baso (×10^9^/L)	0.00 ± 0.00 ^a^	0.00 ± 0.00 ^a^	0.00 ± 0.00 ^a^	0.00 ± 0.00 ^a^	0.00 ± 0.00 ^a^	0.00 ± 0.00 ^a^	0.00 ± 0.00 ^a^
%	0.00 ± 0.00 ^a^	0.00 ± 0.00 ^a^	0.00 ± 0.00 ^a^	0.00 ± 0.00 ^a^	0.00 ± 0.00 ^a^	0.00 ± 0.00 ^a^	0.00 ± 0.00 ^a^
PLT (×10^9^/L)	1038 ± 101.11 ^a,c^	257.8 ± 214.09 ^b^	747.67 ± 184.17 ^a,c^	458.2 ± 246.88 ^b,c^	498 ± 415.7 ^b,c^	77.0 ± 100.58 ^b^	76.5 ± 26.16 ^b^

Data are expressed as mean ± SD (*n* = 4). ^a, b, c^ Different letters within the same row indicate significant differences (*p* < 0.05, one-way ANOVA with LSD post hoc tests). Abbreviation: RBC (red blood cell), Hb (haemoglobin), PCV (packed cell volume), MCV (mean corpuscular volume), MCHC (mean corpuscular haemoglobin concentration), WBC (white blood cell), Band N (band neutrophil), Neutro (Neutrophil), Lymp (Lymphocyte), Mono (monocyte), Eosin (eosinophil), Baso (basophil) and PLT (platelet).

**Table 5 pharmaceutics-14-00649-t005:** Kidney and liver functions of different groups at the end of treatment.

	Normal Control	Untreated Diabetes	Glibenclamide (10 mg/kg)	hUC-MSCs-sEVs (1 mg/kg)	hUC-MSCs-sEVs (0.5 mg/kg)	hUC-MSCs-sEVs (0.25 mg/kg)	hUC-MSCs (2 million Cells)
**Kidney function**							
Na (mmol/L)	137.83 ± 3.18 ^a^	120.8 ± 10.01 ^b^	164.63 ± 17.54 ^c^	133.32 ± 10.22 ^a,b^	126.7 ± 4.81 ^a,b^	122.15 ± 5.44 ^a,b^	142.65 ± 15.06 ^a^
K (mmol/L)	4.72 ± 0.38 ^a^	3.83 ± 0.15 ^b^	5.47 ± 0.841 ^c^	4.69 ± 0.55 ^a^	5.09 ± 0.51 ^a,c^	4.83 ± 0.30 ^a^	4.40 ± 0.52 ^a,b^
Cl (mmol/L)	110.13 ± 2.59 ^a^	89.15 ± 10.6 ^b^	127.97 ± 15.44 ^c^	104.36 ± 7.87 ^a,d^	95.77 ± 3.01 ^b,d,e^	91.5 ± 4.38 ^a,b^	106.7 ± 10.11 ^a,e^
Urea (mmol/L)	4.63 ± 0.91 ^a^	7.27 ± 1.21 ^b^	5.72 ± 0.89 ^a,b^	6.78 ± 2.22 ^b^	8.02 ± 1.77 ^b^	7.18 ± 1.52 ^b^	7.89 ± 1.93 ^b^
Creat (µmol/L)	40.64 ± 3.56 ^a^	42.49 ± 10.62 ^a^	144.58 ± 71.61 ^b^	45.03 ± 1.69 ^a^	47.60 ± 9.72 ^a^	49.71 ± 6.68 ^a^	61.03 ± 7.01 ^a^
**Liver function**							
TP (g/L)	59.58 ± 12.77 ^a^	59.88 ± 26.89 ^a^	72.47 ± 28.44 ^a^	67.98 ± 23.86 ^a^	41.27 ± 3.66 ^a^	70.5 ± 34.89 ^a^	70.63 ± 17.33 ^a^
ALB (g/L)	27.16 ± 2.21 ^a^	26.90 ± 8.68 ^a^	32.02 ± 7.71 ^a^	26.84 ± 6.42 ^a^	22.08 ± 2.54 ^a^	24.58 ± 6.90 ^a^	36.03 ± 6.40 ^a^
Glo (g/L)	32.43 ± 11.56 ^a^	32.99 ± 24.11 ^a^	60.86 ± 1.92 ^a^	41.14 ± 19.25 ^a^	19.19 ± 5.59 ^a^	45.92 ± 33.84 ^a^	34.6 ± 17.90 ^a^
ALB:Glo ratio	0.93 ± 0.31 ^a^	1.02 ± 0.48 ^a^	0.46 ± 0.09 ^a^	0.75 ± 0.31 ^a^	1.24 ± 0.45 ^a^	0.80 ± 0.68 ^a^	1.33 ± 0.89 ^a^
ALP (U/L)	202.20 ± 80.59 ^a^	1094.88 ± 161.59 ^b^	628.43 ± 57.23 ^c^	1076.66 ± 429.57 ^b^	853.24 ± 193.20 ^b,c^	899.09 ± 222.73 ^b,c^	1008.46 ± 90.69 ^b^
AST (U/L)	158.90 ± 36.18 ^a^	123.85 ± 14.45 ^a^	203.91 ± 41.22 ^a^	186.98 ± 76.8 ^a^	167.51 ± 74.54 ^a^	126.79 ± 25.73 ^a^	146.31 ± 23.97 ^a^
ALT (U/L)	53.27 ± 9.50 ^a^	90.01 ± 13.15 ^b^	139.65 ± 31.71 ^c^	105.99 ± 24.72 ^b,c^	124.89 ± 40.83 ^c^	95.48 ± 22.70 ^b^	93.43 ± 27.68 ^b^
DBil (µmol/L)	0.35 ± 0.20 ^a^	2.33 ± 3.47 ^a^	1.44 ± 1.97 ^a^	9.34 ± 18.18 ^a^	3.99 ± 3.01 ^a^	2.24 ± 2.09 ^a^	7.96 ± 15.32 ^a^
TBil (µmol/L)	2.62 ± 1.33 ^a^	4.23 ± 4.30 ^a^	3.61 ± 3.04 ^a^	2.91 ± 1.68 ^a^	5.73 ± 5.02 ^a^	4.61 ± 3.79 ^a^	15.68 ± 27.75 ^a^

Data are expressed as mean ± SD (*n* = 4). ^a, b, c^ Different letters within the same row indicate significant differences (*p* < 0.05, one-way ANOVA with LSD post hoc tests). Abbreviation: Na (sodium), K (potassium), Cl (chloride), Creat (creatinine), TP (total protein), ALB (albumin), Glo (globulin), ALP (alkaline phosphatase), AST (aspartate aminotransferase), ALT (alanine aminotransferase), DBil (direct bilirubin), TBil (total bilirubin).

## Data Availability

Data are contained within the articles.

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
