# Peer review of "Human Umbilical Cord Mesenchymal Stem Cell-Derived Small Extracellular Vesicles Ameliorated Insulin Resistance in Type 2 Diabetes Mellitus Rats"

_pharmaceutics, 2022, doi:10.3390/pharmaceutics14030649_

Round 1

Reviewer 1 Report

The authors report on HUCMSC-derived EVs as a potential anti diabetic therapy. This is an interesting topic, and highly important. The manuscript needs additional work before it can be published. Specific comments are below.

1) Since the EVs hold such a central part in this work, more information (in brief) needs to be given about EV isolation; currently there is only a reference.

2) In Figure 3, panel C is missing

3) In Figure 5, the Y axis label does not correspond to the legend.

4) In some of the results, the effect of the treatment seems minor ( e.g. Fig.7A), or the error bars overlap considerably ( e.g. Fig.7B). This brings doubts at to the significance of these results.

5) Figures 8,9,10 are low-contrast and not clear enough to discern some of the marked features. Also, different panels in Fig. 8 seem to have different resolution, and in all these figures there are no size bars which makes it impossible to ascertain constant magnification.

6) English style needs to be improved throughout. Examples: 

line 40: "protective effects in T2DM rats, which serve as a potential therapy for diabetic patients"

line 336: "Treatment of cytochalasin B with 30 min of incubation was significantly reduced the 2-DG uptake by HSkMCs."

line 506: "treatment group at 0.5showed the degranulation of cytoplasm in the deteriorating cells was occurred"

Reviewer 2 Report

Yap et al. investigated the efficacy of low doses of human umbilical cord MSC-sEVs treatment on improving insulin sensitivity and ameliorating insulin resistance of human skeletal muscle cells (HSkMCs) in vitro and T2DM rat models. Finding is timely and significant. However, the manuscript seems to be immature to be published in Pharmaceutics. The detailed comments are listed below.

  1. Line 103. Heterogeneity and purity of therapeutic EVs are central challenges. Please briefly describe the method for EV isolation.
  2. Line 110. Please add more details of dynamic light scattering measurement, such as EV concentrations. Also, I cannot find how the EV concentrations (protein concentrations?) were measured.
  3. Line 269. Why were p values all indicated as one *? Please add more detailed p-values.
  4. Fig. 1A. The TEM image does not look like a typical EV image. Please show several EVs.
  5. Fig. 1B. Please briefly explain what GRP94 is.
  6. Fig. 1C. There are small and large peaks. What are those? These EV populations may indicate the low purity of the EV sample.
  7. Lines 276-278. Generally, Nanoparticle tracking analysis (NTA) indicates a different measurement from DLS.
  8. Fig. 3B. Please enlarge the image and increase its magnification to see the multinuclear cells. A panel C is missing.
  9. Line 151. Please briefly explain Cytochalasin B (CB) and the rationale for mimicking T2DM in vitro since CB is known for inhibiting F-actin formation.
  10. Line 342. Please briefly explain what nicotinamide is.
  11. Discussion is hard to follow in the current manuscript. Please consider removing too much detail in citing previous studies and briefly describing their finding and conclusion so that readers can easily understand their research background.

Reviewer 3 Report

Human umbilical cord mesenchymal stem cells-derived small extracellular vesicles (hUC-MSCs-sEVs) therapy has shown promising results to treat diabetes mellitus in clinical studies. The hUC-MSCs-sEVs treatment at 1 mg/kg improved glucose tolerance in T2DM rats and showed a protective effect on complete blood count. The authors suggest hUC-MSCs-sEVs could ameliorate insulin resistance and exert protective effects in T2DM rats.

In general, the manuscript contain relevant paragraphs that have been discussed. The selection of bibliography is appropriate to the content of the manuscript. In the conclusion, the authors have included short thoughts. Perhaps even too brief reflections.

Finally, regarding methodology, authors refer about statistics thus the readers can make assumptions regarding the quality and the confidence of the results and reasonability of consideration of the authors. Is such a large standard deviation acceptable in Tables 4 and 5? This raises my question.

The manuscript is very enjoyable to read, but after close evaluation of the paper I suggest revision according to the next point:

  1. Indicate the element of novelty.

  1. Figure 1C: Nanoparticle tracking analysis showed that hUC-MSCs-sEVs are in the range of about 40 to 400 nm with a main population (%) centred at about 144.1 nm (Figure 1C).

Please explain why the following are present in Figure 1C hUC-MSCs-sEVs are in the range of about 4 to 20 nm, and about 1400 to 5000 nm?

I am not a native English speaker, so I don't know if the manuscript would benefit from careful and profound proof-reading and correction of language and style.

Round 2

Reviewer 1 Report

The authors have addressed my prior concerns in the revised manuscript.

Reviewer 2 Report

The authors addressed all of my comments in the response letter and significantly improved the manuscript. Now I recommend the manuscript to be published in Pharmaceutics.